# Global air pollution exposure and poverty

Jun Rentschler [1] ✉ & Nadezda Leonova [1] ✉

Air pollution is one of the leading causes of health complications and mortality worldwide, especially affecting lower-income groups, who tend to be more exposed and vulnerable. This study documents the relationship between ambient air pollution exposure and poverty in 211 countries and territories. Using the World Health Organization's (WHO) 2021 revised fine particulate matter (PM2.5) thresholds, we show that globally, 7.3 billion people are directly exposed to unsafe average annual PM2.5 concentrations, 80 percent of whom live in low- and middle-income countries. Moreover, 716 million of the world's lowest income people (living on less than $1.90 per day) live in areas with unsafe levels of air pollution, especially in Sub-Saharan Africa. Air pollution levels are particularly high in lower-middle-income countries, where economies tend to rely more heavily on polluting industries and technologies. These findings are based on high-resolution air pollution and population maps with global coverage, as well as subnational poverty estimates based on harmonized household surveys.

Air pollution has wide-ranging and profound impacts on human health and well-being. Poor air quality has been shown to be responsible for over 4 million deaths each year from outdoor pollutants, 2.3 million from indoor air pollution[1], and a wide range of cardiovascular, respiratory, and neurological diseases[2–6]. It also impacts productivity, exacerbates inequalities[2], and reduces cognitive abilities[3].

Studies show that the vast majority of the world's population faces unsafe air pollution levels[4,5]. Exposure is especially high in major urban centers, where 86 percent of people live in areas that exceed the WHO's 2005 guideline threshold of 10 µg/m[3][6]. Yet, our understanding of what constitutes unsafe levels of air pollution levels is still evolving. Based on the latest medical evidence, the WHO updated its air quality guidelines in 2021, revising the threshold down to 5 µg/m[3] and significantly increasing the stringency of its 2005 guidelines.

A growing evidence base also highlights the unequal distribution of exposure to and impact of air pollution, with the burden falling disproportionately on lower-income and more marginalized communities[7,8]. The evidence is in strong agreement that air pollution —predominantly the result of human activities—is one of the leading causes of death in low- and middle-income countries[9], where less stringent air quality regulations, the prevalence of older, more polluting machinery and vehicles, fossil fuel subsidies, congested urban transport systems, rapidly developing industrial sectors, and cut-and-burn practices in agriculture all contribute to heightened concentration levels[10].

As health and productivity suffer, evidence from the United States has shown that air pollution reinforces socioeconomic inequalities— with ethnic minorities and low-income populations often exposed to higher pollution levels[11–17]—and that these disparities have increased over time[7]. These groups also tend to be more vulnerable to the impacts of pollution[8], as low-paying jobs are more likely to require physical and outdoor labor, increasing people's exposure. With industrial plants, transport corridors, and other pollution sources disproportionately placed in low-income neighborhoods, air pollution is higher in these areas[7,17,18], driving down housing prices and reinforcing their status as low-income neighborhoods[19,20]. Finally, constraints on healthcare accessibility, availability, and quality further increase air pollution-related mortality among low-income groups[9,21].

Substantial evidence from the United States illustrates how socioeconomic marginalization can increase people's exposure and vulnerability to air pollution, and there are many documented individual cases of environmental inequalities[22]. But there is limited evidence at the global scale on how people's exposure to harmful air pollution interacts with poverty and how this pollution burden is distributed across and within low- and middle-income countries. This is often due to a lack of socioeconomic data with high spatial disaggregation.

A better understanding of the interplay between air pollution and poverty could be crucial for several reasons[23]. Studies from high-income countries on the mortality and morbidity associated with air pollution may not be directly transferable to low-income countries and

[1]The World Bank, Washington, WA, USA. ✉e-mail: jrentschler@worldbank.org; nleonova@worldbank.org

communities, where the nature of occupations and health care differ substantially[24]. The health and productivity implications of unsafe air pollution will also impact low- and middle-income countries' socio-economic development prospects. This is especially pertinent in low-income countries, which—as this study shows—still tend to have relatively low pollution levels compared to more industrialized, middle-income countries. In these countries, it is important to ensure that future development progress does not intensify air pollution, with its associated adverse effects.

Against this context, this study explores the global prevalence of unsafe outdoor air pollution and the way it interacts with poverty (defined as daily expenditure below $1.90, $3.20, and $5.50, respectively, as defined by the World Bank; see Methods). Reflecting 2018 and 2020 conditions, we use global high-resolution data on ambient air pollution (outdoor PM2.5 concentrations), population distribution, and poverty to provide aggregate exposure estimates (see Methods). We show that pollution levels are most hazardous in middle-income countries, where economies tend to rely more heavily on polluting industries and technologies.

Overall, this study contributes to the literature in two ways by offering global estimates of (i) population PM2.5 exposure, based on the WHO's revised air pollution guidelines[25], with detailed national and subnational estimates and (ii) how these interact with national and subnational poverty levels.

## Results

### Global and regional air pollution exposure
Our estimates show that, globally, 7.3 billion people face air pollution levels that are considered unsafe by the WHO—that is, they are exposed to annual average PM2.5 concentrations over $5\,\mu g/m^3$, which increases mortality rates by 4 percent compared to safe areas. Of these, 6.2 billion are directly exposed to at least moderate (over $10\,\mu g/m^3$) air pollution levels and an 8 percent increase in mortality risk, and 2.8 billion are exposed to hazardous (over $35\,\mu g/m^3$) pollution levels and a 24 percent increase in mortality risk. Globally, only 462 million people are exposed to PM2.5 concentrations that are lower than $5\,\mu g/m^3$, the WHO's "safe" threshold (Fig. 1a). Considering a global population of 7.7 billion, this means that approximately 94 percent of the world's population is exposed to unsafe levels of PM2.5 concentration.

Regionally disaggregating global exposure headcounts show that air pollution risks are particularly prevalent in some regions. At 2.2 billion people, East Asia and Pacific (EAP) has the highest number of people exposed to unsafe PM2.5 concentrations, corresponding to about 95 percent of its total population. In South Asia (SAR), about 1.8 billion people (99 percent) are exposed to unsafe air pollution levels. In all other regions, the share of the overall population exposed to unsafe PM2.5 concentrations is smaller, at 92–94 percent in the Middle East and North Africa (MENA), Sub-Saharan Africa (SSA), Europe and Central Asia (ECA), and the United States and Canada (USA & CAN), and 84 percent in Latin America and the Caribbean (Fig. 1b).

### Countries with the largest air pollution-exposed populations
Estimates confirm that several countries stand out with particularly large populations directly exposed to unsafe air pollution levels[26]. The world's two most populous countries—China and India—have the highest absolute population exposure to unsafe air pollution and are home to about 38 percent of all people exposed to unsafe concentrations of PM2.5. In India, 1.36 billion people (99 percent of the population) are exposed to unsafe PM2.5 concentrations (over $5\,\mu g/m^3$); and 1.33 billion (96 percent) to hazardous levels (over $35\,\mu g/m^3$). In China, 1.41 billion people (99 percent of the population) face unsafe PM2.5 concentrations (over $5\,\mu g/m^3$), and 0.765 billion (53 percent) face hazardous levels (Fig. 1c).

Presenting relative exposure estimates for all countries, Fig. 2 demonstrates that in large parts of the world and across all regions, the vast majority of the population is exposed to PM2.5 levels over $5\,\mu g/m^3$. Unlike flood hazards, which are highly localized, air pollution tends to cover and move across large areas, blanketing entire cities or regions. So, if large proportions of a population live in densely populated areas, they tend to be collectively exposed to unsafe pollution levels. Considering a higher pollution threshold of $15\,\mu g/m^3$ shows that populations in low- and middle-income countries—in parts of Central and South America, across Western and Middle Africa, Eastern Europe, Middle East, and Central, South, and East Asia—face high exposure levels (Fig. 2b), while in Eastern China, the Indian subcontinent, and parts of West Africa, large parts of the population face hazardous PM2.5 concentrations (Fig. 2c).

### Poverty and air pollution
Evidence suggests that low-income communities tend to be both disproportionately exposed to unsafe air pollution levels and more vulnerable to serious health impacts[3,27]. Low-income groups tend to be more exposed to air pollution because they are more likely to depend on jobs that require outdoor physical labor, and when affected by pollution-related diseases, they tend to have more limited access to adequate and affordable health care, increasing mortality rates. Low-income countries often also have less developed healthcare systems. So, considering the interplay between pollution, exposure, and poverty can shed light on the vulnerability of affected populations.

Combining air pollution exposure estimates with survey-based subnational poverty data allows us to estimate exposure of the global population living in poverty (Table 1). Our estimates show that 716 million people living on less than $1.90 per day are directly exposed to unsafe PM2.5 concentrations—405 million (57 percent) of them in Sub-Saharan Africa (Fig. 3)—and 275 million are exposed to hazardous PM2.5 concentrations. Countries where poverty and unsafe air pollution coincide also score poorly in terms of health care access and quality, thus exacerbating vulnerabilities (Fig. 3c). Approximately one in every 10 people exposed to unsafe levels of air pollution lives in extreme poverty.

When we use less extreme (i.e., higher) poverty thresholds, the number of air pollution and poverty-exposed people increases significantly. We estimate that around 1.8 billion people living on less than $3.20 a day and 2.9 billion people living on less than $5.50 a day live in unsafe air pollution areas. In Sub-Saharan Africa, increasing the poverty threshold from $1.90 to $5.50 doubles the number of people living in poverty and exposed to unsafe PM2.5 levels from 405 to 877 million (In Sub-Saharan Africa, 39.3 percent of the region's total population lives in extreme poverty ($1.90), and 91.82 percent of the region's total population faces unsafe PM2.5 levels (over 5 μg/m3)). In South Asia and East Asia, it increases more than six-fold, from 220 million to 1.4 billion and 38 to 229 million, respectively. Overall, four in 10 people exposed to unsafe PM2.5 levels live on less than $5.50 a day.

Of the 716 million people living in extreme poverty and exposed to unsafe levels of air pollution, almost half (48.6 percent) are in India, Nigeria, and the Democratic Republic of Congo. With over 202 million, India has the highest number of people living in extreme poverty and exposed to unsafe PM2.5 levels, corresponding to 14.7 percent of its overall population. The 10 countries with the most people who are both living on less than $1.90 a day and exposed to unsafe PM2.5 levels account for 67.8 percent of the world's people exposed to poverty and unsafe PM2.5 concentrations; and seven of the top ten are in Sub-Saharan Africa (Fig. 3b). Although extreme poverty and exposure to unsafe PM2.5 concentrations coincide most acutely in Sub-Saharan Africa, when considering higher poverty thresholds, exposure is also high in the Middle East, South and East Asia, and Latin America (Fig. 4).

### Income and air pollution concentrations
Our estimates on the geographic distribution of PM2.5 exposure suggest that pollution levels differ according to a country's stage of

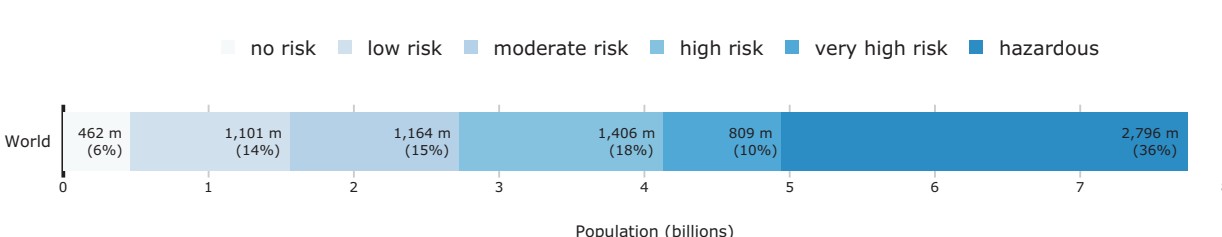

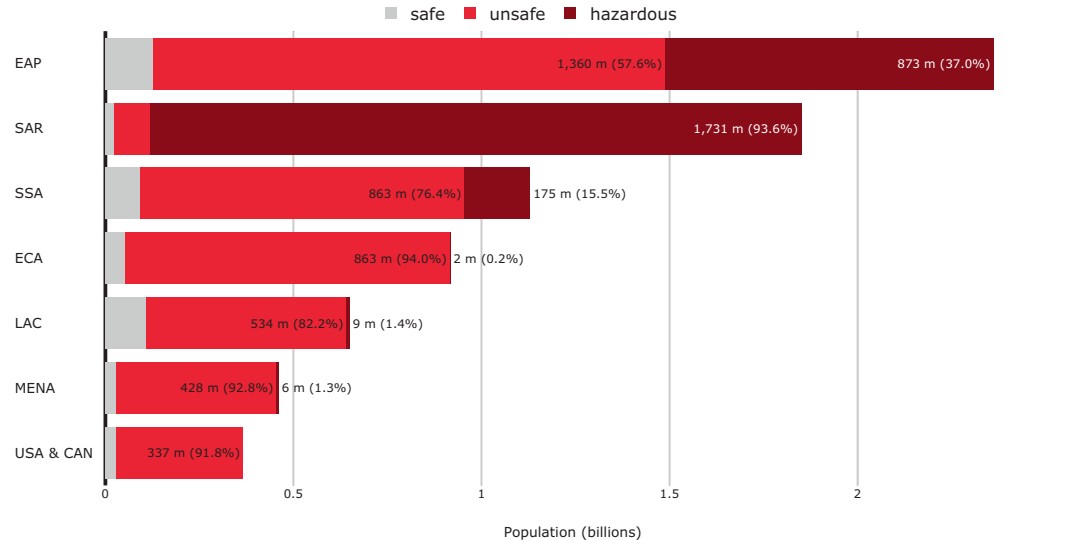

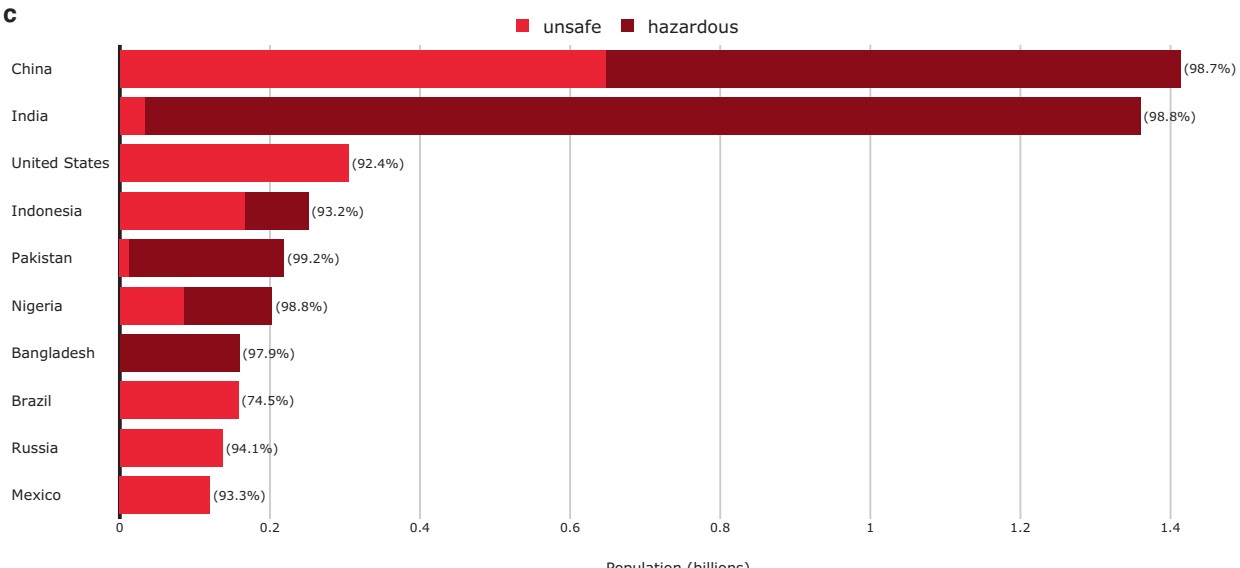

**Fig. 1 | Global population exposure to unsafe air pollution, by region and risk level. a** Global population headcounts exposed to different levels of air pollution risk. **b** Number of people and share of population exposed to air pollution, by region. **c** Top ten countries with highest population exposure to unsafe PM2.5 levels. Hazard categories are defined based on estimated average annual PM2.5 concentration levels. "Unsafe" refers to PM2.5 concentrations over 5 μg/m³. "Hazardous" refers to PM2.5 concentrations over 35 μg/m³.

economic development and industrialization. Most of the people breathing unsafe air live in middle-income countries (Fig. 5). Of the 7.3 billion exposed to unsafe concentrations of PM2.5, 3.4 billion (47.3 percent) live in low- or lower-middle-income countries. Of the 2.8 billion worldwide exposed to hazardous PM2.5 levels, 98.6 percent live in

middle-income countries, compared to just 1.4 percent (40.5 million) in low- and high-income countries combined.

As a share of the overall population, PM2.5 exposure is also highest in lower-middle-income countries (Fig. 5), with about 64.5 percent of people exposed to PM2.5 levels over 35 μg/m³, compared to

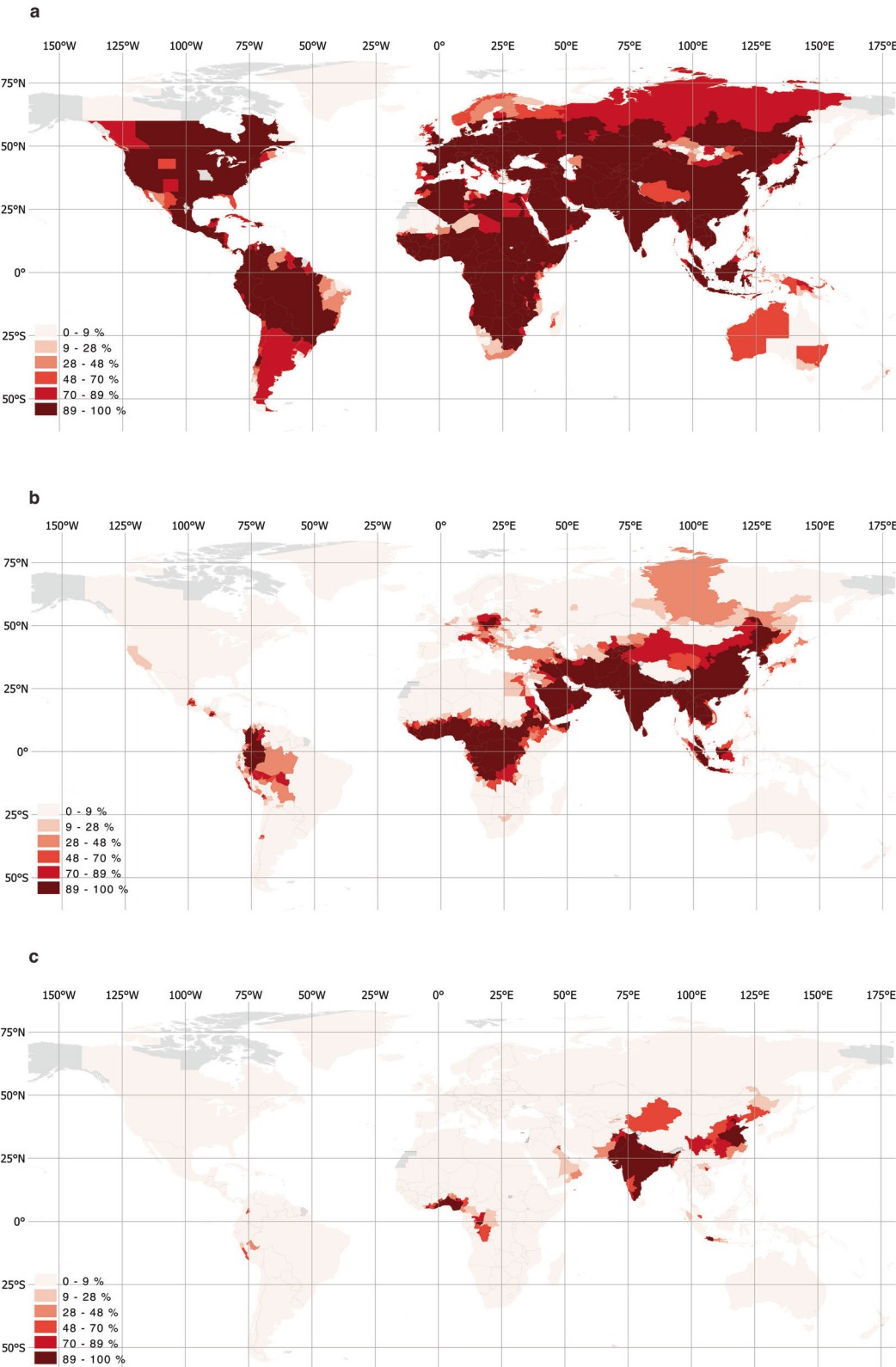

**Fig. 2 | Exposure to unsafe average annual PM2.5 concentrations as a share of the population. a** Percentage of the population exposed to PM2.5 over 5 µg/m. **b** Percentage of population exposed to PM2.5 over 15 µg/m³. **c** Percentage of population exposed to PM2.5 over 35 µg/m³.

just 4.4 percent in low-income countries and 0.9 percent in high-income countries. The pattern holds regardless of which concentration threshold we consider (Fig. 5). The regional distribution of PM2.5 concentrations (Fig. 5d) suggests that these high ambient air pollution levels in middle-income counties are located to a large extent in the countries of South and East Asia, which have experienced rapid economic growth and industrialization in recent decades[6]. Computing spatially averaged PM2.5 concentrations for each of the 2,183 subnational areas in this study and statistically examining their relationship with population and income data also suggests that areas with larger

**Table 1 | PM2.5 exposure headcounts at different poverty thresholds**

| | Poverty threshold (consumption per day) | | |
|---|---|---|---|
| | **$1.90** | **$3.20** | **$5.50** |
| Number of people living in poverty (millions) | 768 | 1,853 | 3,034 |
| Share of the global population that lives in poverty | 9.9% | 23.9% | 39.2% |
| Number of people that live in poverty and are exposed to *unsafe* PM2.5 levels (millions) | 716 | 1,752 | 2,870 |
| Share of the population that lives in poverty and is exposed to *unsafe* PM2.5 levels | 9.3% | 22.6% | 37.1% |
| Number of people that live in poverty and are exposed to *hazardous* PM2.5 levels (millions) | 275 | 938 | 1,573 |
| Share of the population that lives in poverty and is exposed to *hazardous* PM2.5 levels | 3.5% | 12.1% | 20.3% |

Subnational poverty estimates are from the World Bank's Global Subnational Poverty Atlas (see Methods). *Unsafe* levels mean >5 µg/m³; *hazardous* levels mean >35 µg/m³.

populations tend to have higher pollution levels, and average pollution levels appear particularly high for areas in the middle-income category (Supplementary Fig. 3.1).

## Discussion

This study offers a comprehensive account of the relationship between outdoor air pollution exposure, economic development, and poverty in 211 countries and territories. Its global exposure estimates highlight that unsafe air quality poses significant health risks to a vast majority of the global population. We find that 7.3 billion people—that is, 94 percent of the world's population—live in areas that are exposed to PM2.5 concentrations over 5 µg/m³, which increases mortality rates by 4 percent. About 2.8 billion people, or 36 percent of the world population, are directly exposed to concentrations above 35 µg/m³, which increases mortality rates by over 24 percent.

Our study also shows that pollution levels are particularly high in middle-income countries, where a wide range of factors contribute to increased concentration levels. These include less stringent air quality regulations, the prevalence of older, more polluting machinery and vehicles, fossil fuel subsidies, congested urban transport systems, coal-based residential heating, rapidly developing industrial sectors, and cut-and-burn agricultural practices[6,10]. Of the 7.3 billion people exposed to unsafe PM2.5 levels, 80 percent live in low- and middle-income countries. The rapidly growing economies in South and East Asia stand out in terms of absolute exposure, driven by decades of rapid economic growth and industrialization. China (1.41 billion people) and India (1.36 billion) alone account for 38 percent of global exposure to PM2.5 concentrations above WHO guidelines.

This pattern is broadly consistent with the notion of an environmental Kuznets curve, which suggests that air pollution levels would be highest in middle-income countries, where polluting activities, such as manufacturing, dominate the economy while productive capital, such as technology, and regulations tend not to prioritize environmental quality[28,29]. In low-income countries, air pollution concentrations would be relatively low, as economic activities, such as agriculture, tend to rely less on fossil fuels, and the consumption of polluting goods—such as high electricity use or private car ownership—is limited to small population groups. In high-income countries, pollution would be low, as economic activity tends to be focused on less polluting sectors, such as services, polluting activities tend to be offshored, and clean technologies are widely available and mandated by regulation.

Yet these results also imply that the pollution intensity along the economic development path is not set in stone. Whether today's low-income countries indeed witness intensifying pollution as a byproduct of development depends on the availability and affordability of clean technologies, and the incentive structure for adopting them. For example, subsidizing fossil fuel consumption undermines the uptake of clean technologies, entrenching high pollution levels in low- and middle-income countries, where such subsidies are common[30]. Stricter regulations on the embodied pollution content of traded goods can address the offshoring of polluting activities and technologies.

Our study also estimates that 716 million people live in extreme poverty (under $1.90 per day) while facing unsafe air pollution. At least 405 million of them live in Sub-Saharan Africa. Low-income population groups are more likely to perform physical and outdoor labor, and therefore face higher exposure and intake of pollutants. They are particularly vulnerable to prolonged adverse impacts on livelihoods and well-being: with lower access to, and availability and quality of, health care provision, the health risks of exposure to air pollution are probably more severe—and air pollution-related mortality higher—for them than for higher-income households exposed to the same levels. One study on air pollution and infant mortality, for example, suggests that mortality risks in India are two to three times larger than in high-income countries[3]. And, although not covered in this study, exposure to indoor air pollution also affects low-income groups disproportionately, as they tend to be more dependent on polluting, low-cost fuels such as charcoal, kerosene, or firewood for cooking and lighting.

Air pollution is one of the world's leading causes of death, especially affecting lower-income communities, who tend to be more exposed and more vulnerable. Our estimates affirm the case for implementing targeted measures to reduce the pollution intensity of economic growth—for example, by supporting the uptake of less polluting technologies in industry and infrastructure, or facilitating the transition towards cleaner fuels, particularly electrification.

Measures are also warranted to directly address the disproportionate exposure of low-income communities highlighted in this study. Expanding the provision of affordable and adequate healthcare in large urban centers in low- and middle-income countries can help reduce mortality, bringing it closer to levels experienced in higher-income countries. Mandating transparent accounting for environmental and health externalities in planning decisions can help steer pollution sources—such as industrial zones or power plants—away from low-income communities. Finally, removing incentives that perpetuate the over-consumption of fossil fuels can yield a double dividend for lower-income groups. For example, while fossil fuel subsidies confer disproportionate monetary benefits to richer households, the air pollution externalities associated with subsidized fossil fuel consumption are disproportionately borne by low-income households. Addressing such policy distortions can benefit low-income groups in terms of both fiscal and health benefits.

## Methods

This section details the datasets used in this study to calculate global population exposure to high concentrations of air pollution.

### Air pollution data (PM2.5)

Rather than consider the cumulative load of all pollutants, this study looks at the differentiated exposure to anthropogenic PM2.5 pollution across countries. Particulate matter (PM) is one of the most common pollutants, primarily caused by fossil fuel combustion, such as car engines and coal or gas power plants[10]. Airborne PM is commonly categorized by the diameter of particles—PM2.5 for particles of up to

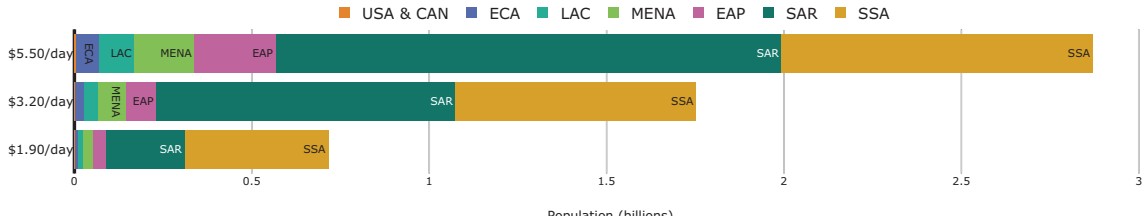

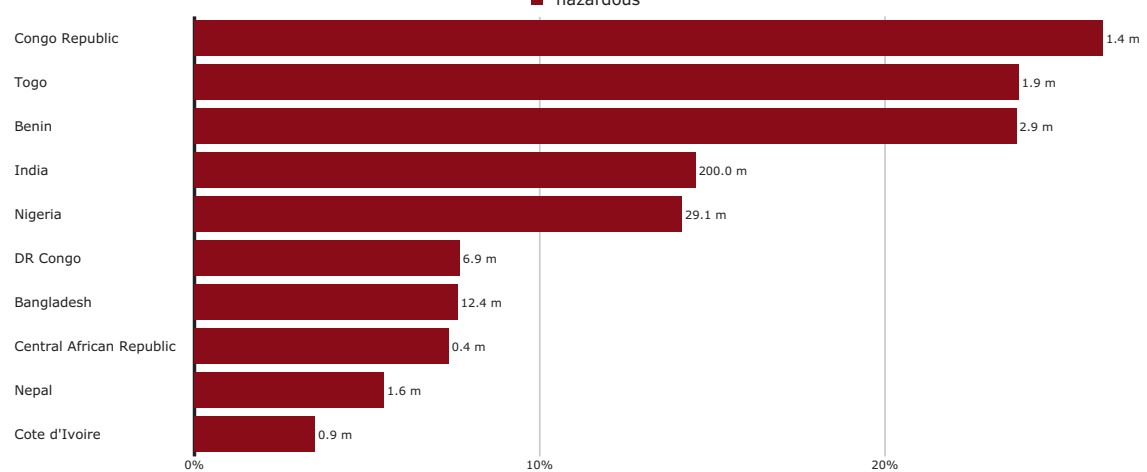

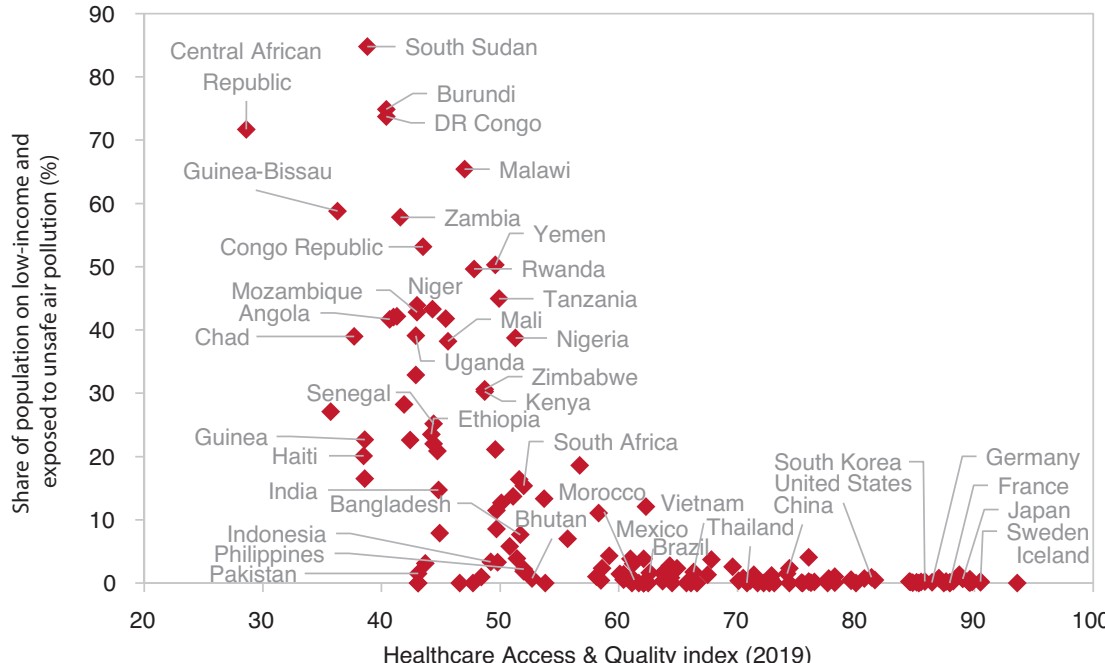

**Fig. 3 | Air pollution exposure of people in poverty. a** Number of people living in poverty and facing unsafe air pollution exposure, at different poverty thresholds and by region. **b** Top ten countries—percentage of people living on $1.90/day and exposed to hazardous PM2.5 levels. **c** Health care access and quality in countries with high air pollution and poverty. The Healthcare Access & Quality (HAQ) index is by GBD 2019 Healthcare Access and Quality Collaborators (2022)[38–41].

2.5 μm in diameter, and PM10 for those up to 10 μm in diameter—as this determines aerial transport, removal processes, and impacts within the respiratory tract[3]. This study focuses on PM2.5, for two main reasons. First, as one of the most pervasive and harmful pollutants, which can

pass through the lungs into the bloodstream and affect other organs, PM2.5 is responsible for the vast majority of air pollution-related deaths, and its impacts are on the rise. It is estimated that 4.5 million people died in 2019 from adverse health effects related to long-term exposure

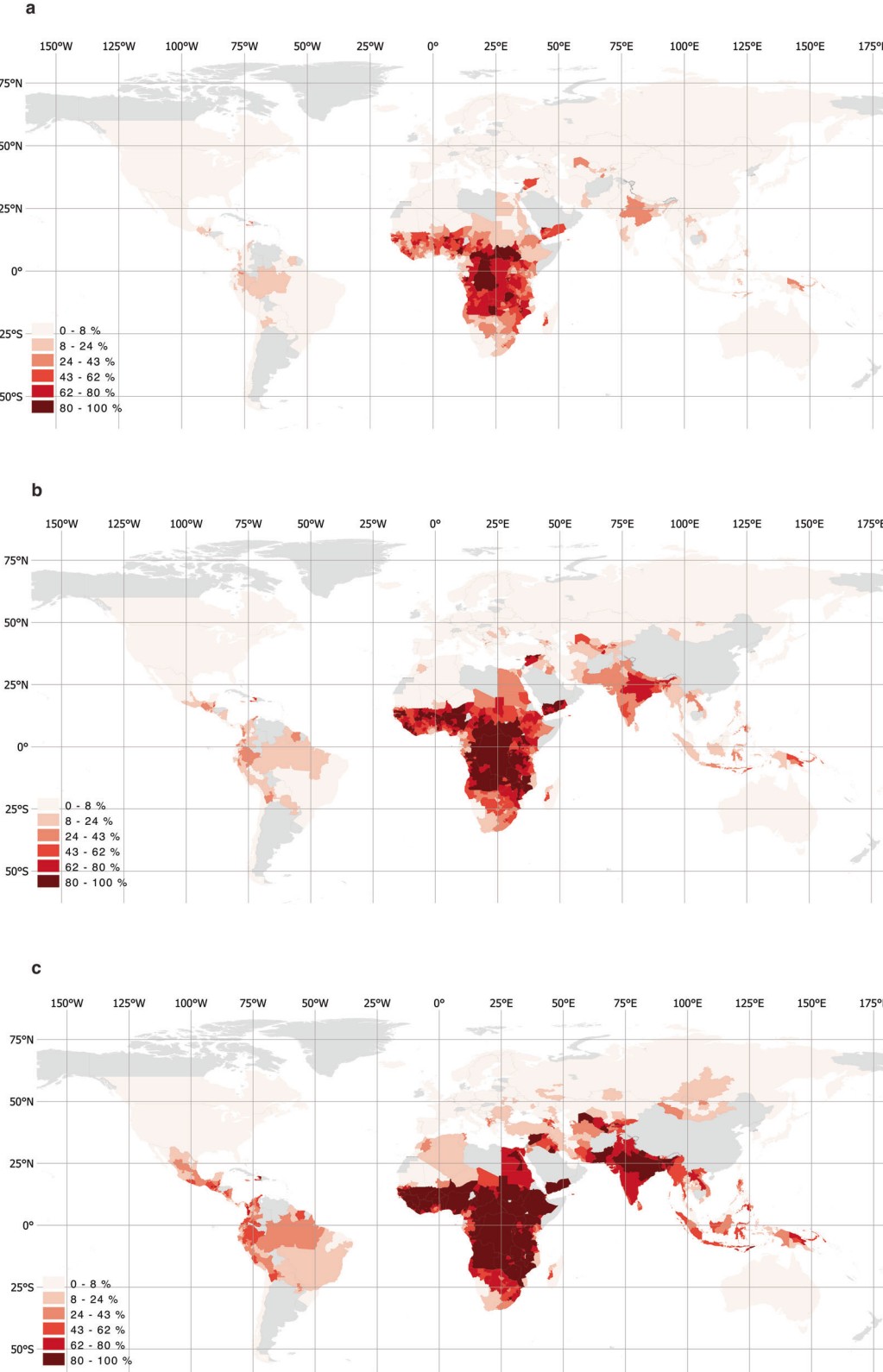

**Fig. 4 | Regional distribution of air pollution and poverty. a** Share of the population exposed to unsafe PM2.5 levels and living on less than $1.90/day. **b** Share of the population exposed to unsafe PM2.5 levels and living on less than $3.20/day. **c** Share of the population exposed to unsafe PM2.5 levels and living on less than $5.50/day.

to ambient air pollution, and that 4.1 million of these deaths were caused by PM2.5 (IHME 2020)[31]. And between 2000 and 2019, PM2.5-attributable deaths increased in all regions except Europe, Latin America, and North America[6]. Second, unlike many other pollutant types, datasets on PM2.5 spatial distribution and concentration levels are available with global coverage. Due to data limitations, this study does not cover indoor air pollution, another pervasive risk to health and well-being, especially in low- and middle-income countries.

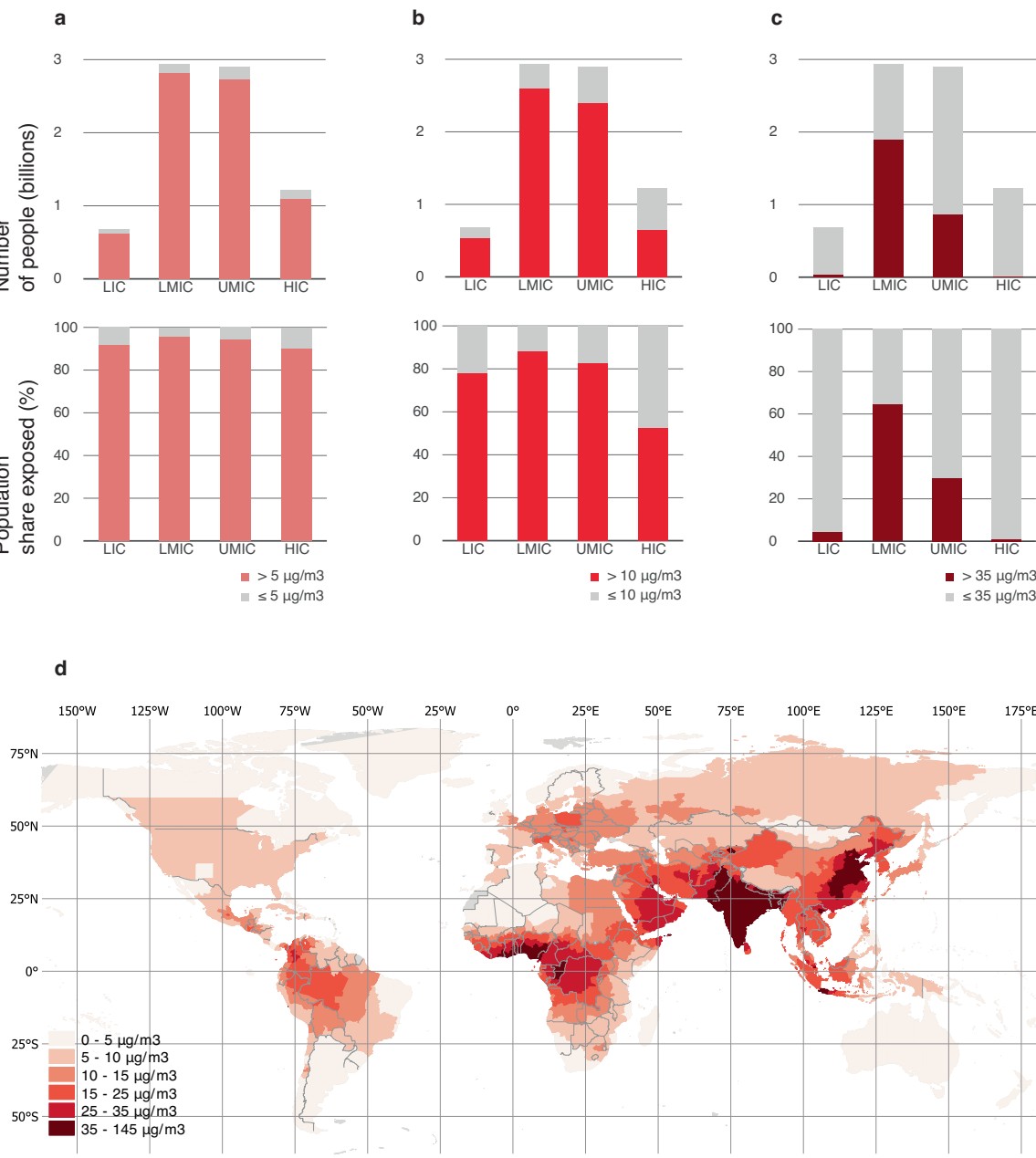

**Fig. 5 | Global population exposure to PM2.5 concentrations. a** Over 5 μg/m³ (4% increased mortality rate). **b** Over 10 μg/m³ (8% increased mortality rate). **c** Over 35 μg/m³ (>24% increased mortality rate). **d** Regional distribution of mean PM2.5 concentrations. Concentration thresholds and estimated mortality rates are based on the WHO Global Air Quality Guidelines[3], which provide details on estimation methods. LIC are low-income countries, LMIC are lower-middle-income countries, UMIC are upper-middle-income countries, HIC are high-income countries.

We use the gridded dataset of ground-level fine particulate matter (PM2.5) concentrations provided by ref. 32, which offers both annual and monthly mean concentrations for 1998–2019, with global coverage and at 0.01-degree resolution (Fig. 6). The dataset is constructed by combining Aerosol Optical Depth satellite retrievals from the NASA MODIS, MISR, and SeaWIFS instruments with the GEOS-Chem chemical transport model, and subsequently calibrating to global ground-based observations using a geographically weighted regression. The 0.01-degree resolution (equivalent to about 1.1 km at the equator) is well suited for capturing regional variation in concentrations, but not granular local variations.

As a globally modeled dataset, some uncertainty is to be expected, though sensitivity tests suggest good agreement with ground measurement[32]. More spatially nuanced analysis—for example, at a neighborhood or street level—would require alternative data based on local measures. It should also be noted that the chemical composition of PM2.5 particles can differ by pollution source[33], and those associated with fossil fuel combustion are more toxic due to higher acidity levels (for example, sulfuric PM from coal burning). The global PM2.5 dataset can inform on total particle concentration, but not on acidity.

**Population data**

To estimate the location of people, we use the WorldPop Global High-Resolution Population dataset, produced by the University of Southampton, the World Bank, and other partners, which offers global coverage and is available yearly from 2000–20. WorldPop provides several datasets, including poverty, demographics, and urban change mapping. This study uses the population count map, a dataset in a raster format, that provides the number of inhabitants per cell, with a 3-arcsecond resolution, thus specifying the distribution of population.

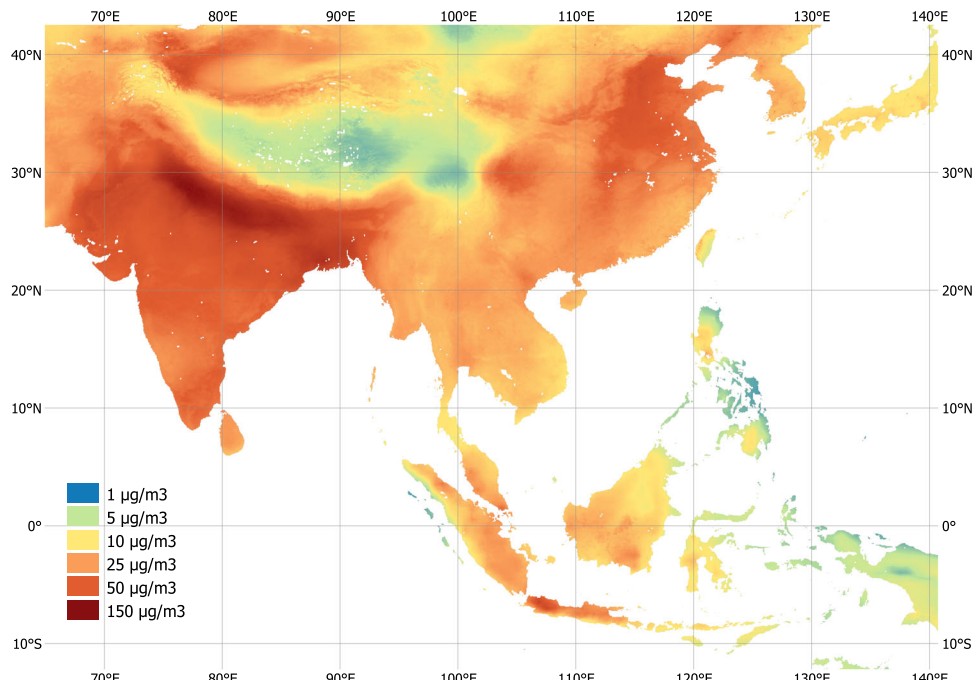

**Fig. 6 | PM2.5 concentrations in Southeast Asia.** Estimates represent annual average concentrations in 2018, constructed based on satellite-based remote sensing data, global chemical transport modeling, and ground measurements. (Source: data by van Donkelaar et al. 2021).

This information is based on administrative or census-based population data, disaggregated to grid cells based on distribution and density of built-up area, which is derived from satellite imagery[34].

The choice of a population density map is important for estimating people's exposure to natural hazards. Smith et al.[35] provide a sensitivity analysis for flood exposure assessments using different population density maps, including WorldPop. They show that high-resolution population density maps perform best in capturing local exposure distribution, particularly the High-Resolution Settlement Layer (HRSL), jointly produced by Facebook, Columbia University, and the World Bank, which has 1-arcsecond or ~30-m resolution. But HRSL is only available for a limited number of countries, and WorldPop is shown to perform better than alternatives with global coverage, such as LandScan data (30-arcsecond, ~900-m resolution)[36].

### Subnational poverty rates

For 1755 of the 2183 subnational units, the World Bank's Global Subnational Poverty Atlas offers poverty estimates, derived from the latest available Living Standards Measurement Survey for the respective country[3]. This harmonized inventory of household surveys offers ground-up empirical poverty estimates. Areas, where no poverty estimates are available tend to be high-income countries and small island states. This study uses the standard World Bank definitions of poverty −that is, daily expenditure thresholds of $1.90, $3.20, and $5.50−to determine the number of people living in poverty in a given subnational administrative unit.

### Administrative boundaries

The definition of national administrative boundaries follows the standard World Bank global administrative map. However, national boundaries are further disaggregated into subnational units for all countries where World Bank household surveys are available with subnational representativeness. These subnational units are typically provinces or states but can also include custom groupings of subnational regions determined by the sampling strategy of household surveys. Overall, this study covers 211 countries, disaggregated into 2183 subnational units.

### Methodology and stepwise computational process

To estimate the number of people exposed to unsafe air pollution levels, this study follows a computational process in four main steps, outlined here.

**Step 1. Resample the PM2.5 data:** First, we resample the air pollution map to ensure that pixels align with the gridded population density map to identify average annual PM2.5 concentration levels along a continuous scale.

**Step 2. Define air pollution risk categories:** Second, we aggregate the values into six risk categories (Table 2), defined in line with the WHO's Air Quality Guidelines[3], which recommend an annual PM2.5 level of up to 5 µg/m³. For countries that exceed this threshold, it recommends interim targets at 10, 15, 25, and 35 µg/m³, corresponding to a linearly increasing mortality rate (Table 2). At higher concentrations, the concentration-response function of mortality may not be linear[37]. For each country, we assign each 1-degree cell one of the six risk categories, repeating this process for the world's landmass of 149 million square kilometers, processing about 300 million data points.

**Step 3. Assign air pollution risk categories to population headcounts at the pixel level and aggregate to the administrative unit:** As the air pollution and population density maps are converted

**Table 2 | PM2.5 concentration thresholds based on the WHO Global Air Quality Guidelines**

| Risk classification | | PM2.5 concentration (µg/m³) | Increased mortality rate (%) |
|---|---|---|---|
| **Safe** | **No/minor risk** | ≤5 | Baseline |
| **Unsafe** | **Low** | 5–10 | 4 |
| | **Moderate** | 10–15 | 8 |
| | **High** | 15–25 | 16 |
| | **Very high** | 25–35 | 24 |
| | **Hazardous** | >35 | >24 |

Concentration thresholds and estimated mortality rates are based on the WHO Global Air Quality Guidelines[3], which provide details on estimation methods.

into the same spatial resolution, we assign each population map cell a unique air pollution risk classification and aggregated them to the administrative unit (such as province or district) level. This allows us to calculate population headcounts for each risk category and for each (sub)national administrative unit, yielding an estimate of the number and share of people exposed to no, low, moderate, high, very high, and hazardous air pollution concentrations throughout the year. Finally, we aggregate these into administrative units—including country and subnational units—to yield regional and global estimates.

**Step 4. Compute the number of people living in poverty and exposed to air pollution risk:** In this final step, we multiply poverty shares with the estimated population headcount exposed to unsafe air pollution, to obtain an estimate of the number of people in each administrative unit living in poverty and exposed to air pollution risk. In the absence of pixel-level poverty share data, we use the World Bank's Global Subnational Poverty Atlas for these calculations, which provide subnational-level data for at least 153 countries.

### Reporting summary
Further information on research design is available in the Nature Portfolio Reporting Summary linked to this article.

### Data availability
Global population count data are provided by WorldPop and publicly available for download at https://hub.worldpop.org/geodata/listing?id=69. Global PM2.5 concentration maps are provided by van Donkelaar et al. (2021) and are publicly available for download at https://sites.wustl.edu/acag/datasets/surface-pm2-5/. Global subnational poverty rate estimates are provided by the World Bank and are publicly available for download at https://datacatalog.worldbank.org/search/dataset/0042041.

### Code availability
The Python source code for this study is available at https://doi.org/10.5281/zenodo.8016653

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

## Acknowledgements

This study has benefited from helpful comments, feedback, and inputs by Mattia Amadio, Esteban Balseca, Samira Barzin, Lander Bosch, Richard Damania, Ira Dorband, Xinming Du, Bramka Jafino, Kichan Kim, Christoph Klaiber, Helena Naber, Jason Russ, Melda Salhab, Ernesto Sanchez-Triana, Lucy Southwood, Margaret Triyana, and Esha Zaveri. The study was supported by the Korea Green Growth Trust Fund.

## Author contributions

J.R. led the study design, analysis, and drafting. N.L. implemented the computational process and contributed to analysis and drafting. All authors critically revised the manuscript and gave final approval for publication.

## Competing interests

The authors declare no competing interests.
