## [Peer Review File · Nature Communications]

Global Air Pollution Exposure and PovertyREVIEWER COMMENTS

Reviewer #1 (Remarks to the Author):

I enjoyed reading the paper "Global Air Pollution and Poverty." The authors use global satellite data linked to population and demographic estimates to show how pollution exposure is dispersed over the world. They find that 7.3 billion people are directly exposed to unsafe average annual PM2.5 concentrations. Low- and middle-income countries account for 80 percent of this exposure. The authors use the best data available to estimate how many people are exposed to unhealthy levels of pollution and their estimates are plausible.

I think this is an important topic, and there is more work to be done in this area. The following are some comments to improve the paper:

1) Unfortunately, the contribution of this paper seems to be relatively minimal given the other work on this topic and the corresponding findings in the WHO report (2021). While the authors use satellite data rather than ground monitoring data, the conclusions are much the same as the previous literature on the topic. See, for example, Marcantonio, Javeline, Field and Fuentes (2021) and the WHO report (2021).

2) The authors use a very low threshold of 5 ug/m³ of PM2.5 as the threshold above which air pollution is unsafe with little evidence to support this threshold. Whereas the WHO report (2021) make use of many studies to support their recommendations, the literature review and documentation in this paper is sparse.

3) Similarly, the authors have numbers for "increased mortality rates" for difference PM2.5 concentrations in Figure 5 and Table 3, but there is no documentation from what studies these numbers came from. More documentation and confidence intervals are necessary for the claims being made.

4) Has the distribution of pollution changed over time? Do mortality results vary between countries?

Reviewer #2 (Remarks to the Author):

Referee Report

Global Air Pollution and Poverty

The paper proposes to investigate the interplay between air pollution and poverty at global level. Using global high-resolution data on ambient air pollution (outdoor concentrations of PM2.5), population distribution, and poverty, the authors show that pollution levels are most hazardous in middle-income countries, where economies tend to rely more heavily on polluting industries and technologies. I think the paper proposes an interesting, though challenging, question. However, some further steps could be taken to clarify the argument and enhance the analysis. I hereby list some comments hoping they will be of help to the author/s.

Introduction

The objective of the paper is expressed explicitly in both abstract and introduction. However, I would suggest underlining, already in the introductory section, the novelty/originality of your contribution in relation to the relevant literature.

Literature Review

I would emphasize a more critically what are the underlying research gaps, and which is the novelty of both your contribution and your methodological approach.

Data

The choice of the PM2.5 index should be more strongly and carefully motivated, also explaining more convincingly why the other main local air pollutants held responsible for impacting human health are not explicitly considered. The European Environment Agency (2018) states, in fact, that amongst the main anthropogenic emissions responsible for the quality of the air and the most important pollutants in terms of potential risk for human health, there are the following four main local pollutants: nitrogen monoxide, carbon monoxide, volatile organic compounds, PM10. In this way, the analysis could certainly become much more representative of the global air pollution problem.

Moreover: what is the time-frame of the analysis? Not clear.

Methodology

In economic papers, the "Methods" sections are usually presented and discussed before the "Results" sections; I would consider restructuring the overall manuscript accordingly to this scientific praxis. I'm afraid that here is no sufficient effort to bring the topic under an economic theoretical framework. The paper doesn't really start with a review of the literature on the link between air pollution and economic growth; the Environmental Kuznets Curve as a generally applicable framework for explaining environmental outcomes in general is not even mentioned.

Moreover, methodologically, the paper does not seem to be particularly innovative as it does not employ any econometric regression analysis but relies only on statistical elaborations.

Other comments:

- Line 21: check grammar of the sentence starting with "It also impacts..."
- Some statements and arguments are missing references as listed below. Please provide references for sentences ending in lines: 22 (page 1) with regard to "Studies (please cite some) show that..."; 41 (page 1) with regard to "Industrial plants, transport corridors, and other pollution sources are disproportionately placed in low-income neighborhoods..."; 49 (page 2) with regard to "While most studies have focused on air pollution in rich countries"...; 50 (page 2) with regard to "Studies from high-income countries on the mortality and morbidity associated with air pollution"...; 122 (page 6) with regard to "It is well documented (please cite relevant literature here) that,..."; 224 (page 12) with regard to "For example, one study on air pollution and infant mortality (please cite it) suggests that..."; 248 (page 12) with regard to "PM is one of the most common pollutants, ..." (please cite most relevant literature);
- Line 45: with regard to the statement "there is little evidence documenting the global scale of poor people's exposure to harmful air pollution, and how their pollution burden is distributed across and within low- and middle-income countries", I'm not sure there isn't evidence on this issue. It might be useful to check, for instance, The Environmental Justice Atlas - which documents global social conflicts around environmental issues (<https://ejatlas.org>) - and all the environmental justice literature that relies on this source;
- Lines 62/63: it could be beneficial to provide some motivations on why indoor air pollution is not covered in the analysis;
- Lines 95/102/130/131/177/194/216/228: rather than estimates, I would speak of statistical elaborations;
- Line 130: the survey-based subnational poverty data should be more carefully/more rigorously explained;
- Lines 186-188: the proposed motivations (i.e., "rapid economic growth and industrialization of South and East Asia") should be more carefully/more critically discussed;
- Lines 201-204: it is necessary to provide more rigorous explanations (i.e., what kind of data are these? how were they collected? what years are they referring to?) with regard to "less stringent air quality regulations, the prevalence of older polluting machinery and vehicles, subsidized fossil fuels, congested urban transport systems, rapidly developing industrial sectors, and cut-and-burn agricultural practices";

- Several pieces of the same arguments are spread across different paragraphs. I would invest a bit of time in re-organizing the paper using topic-sentences and making the paragraphs less repetitive;
 - Adding the overall descriptive statistics of the data used in the analysis could be beneficial.
- Moreover, generally, before the table of the Descriptive Statistics, a table called "Variable description and data sources" should be provided.
- Reference list does not follow a systematic/coherent approach.
 - Improving the conclusions with some stronger policy implications would be beneficial.

European Environment Agency – EEA (2018). Air quality in Europe – 2018 report, EEA, Copenhagen, <https://www.eea.europa.eu/publications/air-quality-in-europe-2018>.

Another reference to look at is the following: https://environment.ec.europa.eu/news/european-health-burden-attributable-air-pollution-fell-over-three-decades-1990-2019-2023-01-11_en

Review comments and responses

Nature Communications manuscript NCOMMS-21-40533

("Global Air Pollution and Poverty")

Reviewer 1	
I enjoyed reading the paper "Global Air Pollution and Poverty." The authors use global satellite data linked to population and demographic estimates to show how pollution exposure is dispersed over the world. They find that 7.3 billion people are directly exposed to unsafe average annual PM2.5 concentrations. Low- and middle-income countries account for 80 percent of this exposure. The authors use the best data available to estimate how many people are exposed to unhealthy levels of pollution and their estimates are plausible. I think this is an important topic, and there is more work to be done in this area. The following are some comments to improve the paper:	Thank you for your supportive and constructive comments on our paper, "Global Air Pollution and Poverty." We appreciate your feedback and are glad to hear that you found our research to be important. Your comments were very useful for further refining our study. We fully agree that there is more work to be done in this area, e.g. to better understand vulnerabilities of different population groups in low-income countries, and hope that this study will contribute to this effort.
1) Unfortunately, the contribution of this paper seems to be relatively minimal given the other work on this topic and the corresponding findings in the WHO report (2021). While the authors use satellite data rather than ground monitoring data, the conclusions are much the same as the previous literature on the topic. See, for example, Marcantonio, Javeline, Field and Fuentes (2021) and the WHO report (2021).	Thank you for this feedback. A key contribution of our paper is to estimate the number of poor people who face unsafe air pollution – this has not been done before, and especially not with high resolution data as in our study. While several studies have assessed global air pollution exposure, they have omitted considering poverty or income levels. Yet, this consideration is crucial, as poor people tend to be more exposed to air pollution and more vulnerable to its impacts – not least due to limited access to affordable health care. While we agree that our paper builds on previous work – including by the WHO (2021) report and Marcantonio, Javeline, Field and Fuentes (2021) – we believe that our use of satellite data and subnational empirical poverty data provides a valuable contribution to the field. We have referenced these studies in our paper. However, the relationship between poverty and air pollution exposure was not addressed by past studies. There is no study that provides an estimate of the number of

	poor people exposed to unsafe air pollution, and where they are located. Hence, we believe that our methodology and data analysis provide a unique new perspective that can supplement and strengthen the existing body of literature.
2) The authors use a very low threshold of 5 ug/m³ of PM_{2.5} as the threshold above which air pollution is unsafe with little evidence to support this threshold. Whereas the WHO report (2021) make use of many studies to support their recommendations, the literature review and documentation in this paper is sparse.	Thank you, this is an important point. The WHO recommends a threshold 5 ug/m³ in its air quality guidelines (revised in 2021) – this is why we use 5 ug/m³. The WHO also recommends several interim thresholds of 10, 15, 25 and 35 ug/m³. We agree with the reviewer that 5 ug/m³ is low. This is why we consider all of the WHO’s interim targets (in addition to the 5 ug/m³ threshold). Our results distinguish different threshold levels (e.g. Figure 1, Table 2). We agree with the reviewer that instead of choosing our own threshold (which would risk being arbitrary), it is important to use the WHO’s recommended threshold level. Indeed, the WHO’s threshold is based on an extensive review of the scientific literature. Rather than repeat the literature review conducted by the WHO report (2021), we provide a summary and citations for the main underlying studies. Please note that the journal’s formatting guidelines do not allow for an extensive literature review to be included in our study. In short, our study builds on the WHO (2021) guideline report and we have used the WHO’s recommendations to inform the thresholds used in our analysis.
3) Similarly, the authors have numbers for “increased mortality rates” for difference PM_{2.5} concentrations in Figure 5 and Table 3, but there is no documentation from what studies these numbers came from. More documentation and confidence intervals are necessary for the claims being made.	Thank you for catching this omission. Concentration thresholds and estimated mortality rates are obtained from the WHO Global Air Quality Guidelines report (WHO, 2021), which provide details on estimation methods. The numbers are not claimed by our study, but provided by the WHO. We have clarified this source in the manuscript, including for Figure 5 and Table 2 (formerly labeled Table 3).
4) Has the distribution of pollution changed over time? Do mortality results vary between countries?	Thank you, these are two important points, which we discuss in the study. In response to each point:

	 - Strictly speaking, our study is cross-sectional and therefore does not examine changes in pollution distribution over time. However, it does show that low-income countries tend to have lower pollution concentrations than middle income countries – this may indicate that over time, as countries develop economically, they tend to increase polluting economic activities. However, this path is not set in stone, and wide-scale adoption of clean technologies can support clean development. We have ensured that these arguments are discussed in more detail in the Discussion section. - The mortality rates provided by the WHO (2021) are global and hence they do not vary between countries. This is an important shortcoming of the current literature. Case study evidence suggests that people in low-income communities may face significantly higher mortality rates than rich communities (e.g. due to higher work-related pollution exposure, or limited healthcare access). However, the literature has focused predominantly on mortality studies for the US and other high-income countries. Hence there is no global inventory of country-disaggregated mortality rates for air pollutants, and especially not for low-income countries. Without our study we raise attention to the fact that 716 million poor people face unsafe air pollution – that it is important to also consider the health and livelihoods impacts on these groups.
--	--

Reviewer 2

The paper proposes to investigate the interplay between air pollution and poverty at global level. Using global high-resolution data on ambient air pollution (outdoor concentrations of PM2.5), population distribution, and poverty, the authors	Thank you for your thoughtful and supportive comments on the paper. Your suggestions have helped us to strengthen the manuscript and to clarify the arguments. We provide more detailed responses to your points below. Thank
---	--

show that pollution levels are most hazardous in middle-income countries, where economies tend to rely more heavily on polluting industries and technologies. I think the paper proposes an interesting, though challenging, question. However, some further steps could be taken to clarify the argument and enhance the analysis. I hereby list some comments hoping they will be of help to the author/s.	you for taking the time to provide this feedback.
Introduction The objective of the paper is expressed explicitly in both abstract and introduction. However, I would suggest underlining, already in the introductory section, the novelty/originality of your contribution in relation to the relevant literature.	We agree the literature, its gaps, and our contribution could have been explained more explicitly. We have revisited the introduction section to improve the summary of existing evidence, added several additional references, and ensured that we clearly explain our contribution (towards the end of the introduction section).
Literature Review I would emphasize a more critically what are the underlying research gaps, and which is the novelty of both your contribution and your methodological approach.	Thanks for highlighting this. We're limited by the journal's editorial guidelines which do not allow for a comprehensive stand-alone literature review section like in other journals. Instead we have tried to strengthen the Introduction section to better summarize the literature and our contribution. In short, a lot of the literature on "environmental justice" and inequality in pollution exposure has focused on the US. As you rightly point out below there are many case studies, which we have tried to better reflect in the references. The Environmental Justice Atlas is particularly useful, as it collects many such cases in one platform. However, it also does not provide estimates of the global number of poor people exposed to unsafe air pollution. In line with your comment we have significantly expanded the citations of relevant studies, The introduction section closes by highlighting the contribution of this study in relation to this literature.

Data The choice of the PM2.5 index should be more strongly and carefully motivated, also explaining more convincingly why the other main local air pollutants held responsible for impacting human health are not explicitly considered. The European Environment Agency (2018) states, in fact, that amongst the main anthropogenic emissions responsible for the quality of the air and the most important pollutants in terms of potential risk for human health, there are the following four main local pollutants: nitrogen monoxide, carbon monoxide, volatile organic compounds, PM10. In this way, the analysis could certainly become much more representative of the global air pollution problem. Moreover: what is the time-frame of the analysis? Not clear.	Thank you, we agree the choice of PM2.5 should be more clearly explained. We have revised the Methods sections to clarify this issue. Specifically, we explain that this study focuses on PM2.5, for two main reasons:  (i) PM2.5 is responsible for the vast majority of air pollution–related deaths, and its impacts are on the rise. It is estimated that 4.5 million people died in 2019 from adverse health effects related to long-term exposure to ambient air pollution, with 4.1 million of these deaths caused by PM2.5 (IHME 2020). While it is true that other pollutants can also be very harmful, PM2.5 is by far the most pervasive (i.e. most widely spread). In sheer numbers, PM2.5 is by far the deadliest, as it can pass through the lungs into the bloodstream and affect other organs. (ii) Datasets on the spatial distribution and concentration levels of PM2.5 are available with global coverage, unlike for many other pollutant types. Regarding the time-frame of the analysis, we have also clarified that the study is cross-sectional. IHME (Institute for Health Metrics and Evaluation). 2020. Global Burden of Disease Study 2019 (GBD 2019) Results. Seattle, WA: IHME.
Methodology In economic papers, the “Methods” sections are usually presented and discussed before the “Results” sections; I would consider restructuring the overall manuscript accordingly to this scientific praxis. I’m afraid that here is no sufficient effort to bring the topic under an economic theoretical framework. The paper doesn’t really start with a review of the literature on the link between air pollution and economic growth; the Environmental Kuznets Curve as a generally applicable framework	Thank you for the feedback on the methodology section and structure of the paper. Please note that the structure of the manuscript follows the editorial guidelines of the journal. Hence we believe we’re unable to restructure the manuscript as suggested and would like to defer this decision to the editor for further consideration.

for explaining environmental outcomes in general is not even mentioned. Moreover, methodologically, the paper does not seem to be particularly innovative as it does not employ any econometric regression analysis but relies only on statistical elaborations.	We also took your suggestions regarding the economic theoretical framework into account and made sure to review the literature on the link between air pollution and economic growth, including the Environmental Kuznets Curve (EKC). We have introduced the EKC notion in the discussion section, noting that our results (e.g. Figure 5) are consistent with a EKC type relationship. We note that even though a EKC relationship appears present in this cross-section data, today's low-income countries do not necessarily have to follow the EKC trajectory as "leapfrogging" is possible. Regarding the methodology, we agree that the main results are obtained through geospatial data analysis rather than econometric regressions. However, the high-resolution and global nature of these datasets makes the computational process non-trivial. We revisited explanations in the Methods section to clarify the process. In addition, we note that we have estimated 6 polynomial regression models and 2 non-parametric kernel regressions, which explore the relationship between air pollution levels and income, while controlling for a range of control variables. These results are included in Annex 3 of the Supplementary Material. We have included a cross-reference to these results in the main manuscript (since these regression results back up the main findings presented in Figure 5).
Other comments: - Line 21: check grammar of the sentence starting with "It also impacts..."	Thank you for picking this up. The sentence has been corrected.
- Some statements and arguments are missing references as listed below. Please provide references for sentences ending in lines: 22 (page 1) with regard to "Studies (please cite some) show that..."; 41 (page 1) with regard to "Industrial plants, transport corridors, and other pollution sources are disproportionately placed in low-income neighborhoods..."; 49 (page 2) with regard to "While most studies have focused on air pollution in rich countries"...	Thank you for highlighting these parts. We fully agree that references are useful to back up these statements. We have added relevant citations in these paragraphs. We have also reviewed the rest of the manuscript and added citations in several other places to ensure completeness.

50 (page 2) with regard to “Studies from high-income countries on the mortality and morbidity associated with air pollution”...; 122 (page 6) with regard to “It is well documented (please cite relevant literature here) that,...”; 224 (page 12) with regard to “For example, one study on air pollution and infant mortality (please cite it) suggests that...”; 248 (page 12) with regard to “PM is one of the most common pollutants, ...” (please cite most relevant literature);	
- Line 45: with regard to the statement “there is little evidence documenting the global scale of poor people’s exposure to harmful air pollution, and how their pollution burden is distributed across and within low- and middle-income countries”, I’m not sure there isn't evidence on this issue. It might be useful to check, for instance, The Environmental Justice Atlas - which documents global social conflicts around environmental issues (https://ejatlas.org) – and all the environmental justice literature that relies on this source;	Thanks, this point is well taken. We have modified the language to avoid suggesting that there is no evidence at all. What we mean is that there is no study that estimates the number of poor people exposed to unsafe air pollution globally, and that systematically (globally) assesses the interaction between poverty and pollution concentration levels. However, indeed there are various individual studies focusing on certain locations – e.g. China or India. We have cited these studies in the manuscript now. We also now cite the Environmental Justice Atlas, which is a very useful resource, but we note that it provides a collection of individual cases of environmental conflict – it does not provide a global number of poor people facing high pollution.
- Lines 62/63: it could be beneficial to provide some motivations on why indoor air pollution is not covered in the analysis;	We have clarified that indoor air pollution is not included due to data limitations. To our knowledge, no dataset exists on indoor air quality levels with global coverage. We have clarified that this study focuses on outdoor air pollution, as is common practice in the literature on ambient air quality.
- Lines 95/102/130/131/177/194/216/228: rather than estimates, I would speak of statistical elaborations;	We have clarified in the methods section that our use of the term “estimates” refers to the results from our 4-step computational estimation process. We believe it is important to highlight that our results are “estimates” – i.e. approximations of the true unobservable numbers. We do not want to suggest a false sense of accuracy. However, the description of the 4-step computational process should clarify that results are not based on regression estimation, but instead on processing of geo-spatial data. We would welcome the reviewer’s further feedback on this.

- Line 130: the survey-based subnational poverty data should be more carefully/more rigorously explained;	Thanks for catching this. We have ensured that the poverty dataset is explained in full in the Methods section. We have also included a cross reference under Table 1.
- Lines 186-188: the proposed motivations (i.e., “rapid economic growth and industrialization of South and East Asia”) should be more carefully/more critically discussed;	Thank you for catching this imprecise formulation. The sentence suggested that economic growth and industrialization explain high pollution concentrations, but did not offer evidence for this claim. We have rephrased this sentence in a way that merely highlights the association, and also provided a reference to back up the observation.
- Lines 201-204: it is necessary to provide more rigorous explanations (i.e., what kind of data are these? how were they collected? what years are they referring to?) with regard to “less stringent air quality regulations, the prevalence of older polluting machinery and vehicles, subsidized fossil fuels, congested urban transport systems, rapidly developing industrial sectors, and cut-and-burn agricultural practices”;	Thanks for catching this – we agree. Our study identifies high pollution levels in middle income countries, but does not allow identifying the causes for this. We have added references to other studies which have explored this issue, and ensured that the wording does not suggest that this is explored in our study. We have also supplemented this section with discussion of the environmental Kuznets curve, as suggested in your other comment – this helps clarify why pollution may first rise and then fall with income.
- Several pieces of the same arguments are spread across different paragraphs. I would invest a bit of time in re-organizing the paper using topic-sentences and making the paragraphs less repetitive;	We fully agree. We have revisited the manuscript and streamlined the text and narrative flow. Indeed, we found several repetitions.
- Adding the overall descriptive statistics of the data used in the analysis could be beneficial. Moreover, generally, before the table of the Descriptive Statistics, a table called “Variable description and data sources” should be provided.	While we agree, we are mindful of the strict wordcount and length requirements by the journal. We have added a link to the original study by van Donkelaar (2021) which produced the air pollution data and provides extensive descriptive statistics. Given space constraints we would prefer not to duplicate these descriptive statistics from their study, but are open to further suggestions.
- Reference list does not follow a systematic/coherent approach.	The reference list organizes references in order of appearance in the text. We believe this is in line with the journal’s editorial guidelines, but we welcome the editor’s guidance on this.
- Improving the conclusions with some stronger policy implications would be beneficial.	Thanks. Please refer to the revised Discussion section. We believe there are two key policy implications that emerge:  - “Dirty growth” is not predestined. Air pollution concentrations are highest in

	middle-income countries, and much lower in high-income countries. For today's low-income countries (where air pollution is still relatively low) there is still an opportunity to leapfrog to clean, modern technologies, i.e. avoid an environmental Kuznets type trajectory.  - There are a substantial number of very poor people who already face significant air pollution levels, while also having very limited access to quality healthcare. To reduce mortality, a key priority will be to improve healthcare access and control further pollution growth.
European Environment Agency – EEA (2018). Air quality in Europe – 2018 report, EEA, Copenhagen, https://www.eea.europa.eu/publications/air-quality-in-europe-2018.	
Another reference to look at is the following: https://environment.ec.europa.eu/news/european-health-burden-attributable-air-pollution-fell-over-three-decades-1990-2019-2023-01-11_en	

REVIEWERS' COMMENTS

Reviewer #2 (Remarks to the Author):

The authors have carefully followed the comments and suggestions provided, and the round of changes has improved the clarity and the organization of the manuscript. Overall, the contribution to the literature on the relationship between air pollution and poverty is relevant and timely. I, therefore, recommend the article for publication.